# Counterfactually Comparing Abstaining Classifiers

**Yo Joong Choe**[*]
Data Science Institute
University of Chicago
yjchoe@uchicago.edu

**Aditya Gangrade**
Department of EECS
University of Michigan
aditg@umich.edu

**Aaditya Ramdas**
Dept. of Statistics and Data Science
Machine Learning Department
Carnegie Mellon University
aramdas@cmu.edu

## Abstract

Abstaining classifiers have the option to abstain from making predictions on inputs that they are unsure about. These classifiers are becoming increasingly popular in high-stakes decision-making problems, as they can withhold uncertain predictions to improve their reliability and safety. When *evaluating* black-box abstaining classifier(s), however, we lack a principled approach that accounts for what the classifier would have predicted on its abstentions. These missing predictions matter when they can eventually be utilized, either directly or as a backup option in a failure mode. In this paper, we introduce a novel approach and perspective to the problem of evaluating and comparing abstaining classifiers by treating abstentions as *missing data*. Our evaluation approach is centered around defining the *counterfactual score* of an abstaining classifier, defined as the expected performance of the classifier had it not been allowed to abstain. We specify the conditions under which the counterfactual score is identifiable: if the abstentions are stochastic, and if the evaluation data is independent of the training data (ensuring that the predictions are *missing at random*), then the score is identifiable. Note that, if abstentions are deterministic, then the score is unidentifiable because the classifier can perform arbitrarily poorly on its abstentions. Leveraging tools from observational causal inference, we then develop nonparametric and doubly robust methods to efficiently estimate this quantity under identification. Our approach is examined in both simulated and real data experiments.

## 1 Introduction

Abstaining classifiers (Chow, 1957; El-Yaniv and Wiener, 2010), also known as selective classifiers or classifiers with a reject option, are classifiers that have the option to abstain from making predictions on certain inputs. As their use continues to grow in safety-critical applications, such as medical imaging and autonomous driving, it is natural to ask how a practitioner should *evaluate and compare* the predictive performance of abstaining classifiers under black-box access to their decisions.

In this paper, we introduce the *counterfactual score* as a new evaluation metric for black-box abstaining classifiers. The counterfactual score is defined as the expected score of an abstaining classifier's predictions, *had it not been allowed to abstain*. This score is of intrinsic importance when the potential predictions on abstaining inputs are relevant. We proceed with an illustrative example:

**Example 1.1** (Free-trial ML APIs). Suppose we compare different image classification APIs. Each API has two versions: a free version that abstains, and a paid one that does not. Before paying for the full service, the user can query the free version for up to $n$ predictions on a user-provided dataset, although it may choose to reject any input that it deems as requiring the paid service. Given two such APIs, how can the practitioner determine which of the two paid (non-abstaining) versions would be better on the population data source, given their abstaining predictions on a sample?

---

[*]This work was submitted while this author was at Carnegie Mellon University.

37th Conference on Neural Information Processing Systems (NeurIPS 2023).

Example 1.1 exhibits why someone using a black-box abstaining classifier would be interested in its counterfactual score: the user may want to reward classifiers whose hidden predictions are also (reasonably) good, as those predictions may be utilized in the future. This is a hypothetical example, but we can imagine other applications of abstaining classifiers where the counterfactual score is meaningful. In Appendix A.1, we give three additional examples, including safety-critical applications where the hidden predictions may be used as a backup option in a failure mode.

To formally define, identify, and estimate the counterfactual score, we cast the evaluation problem in Rubin (1976)'s missing data framework and treat abstentions as *missing predictions*. This novel viewpoint directly yields nonparametric methods for estimating the counterfactual score of an abstaining classifier, drawing upon methods for causal inference in observational studies (Rubin, 1974; Robins et al., 1994; van der Vaart, 2000), and represents an interesting yet previously unutilized theoretical connection between selective classification, model evaluation, and causal inference.

The identification of the counterfactual score is guaranteed under two standard assumptions: the missing at random (MAR) condition, which is satisfied as long as the evaluation data is independent of the classifier (or its training data), and the positivity condition. As with standard missing data settings, both the MAR and positivity conditions are necessary for identification. We later discuss each condition in detail, including when the positivity condition is met and how a policy-level approach may be necessary for safety-critical applications.

The counterfactual score can be viewed as an alternative to the selective score (mean score on nonabstentions) and the coverage (1 minus the abstention rate) (El-Yaniv and Wiener, 2010), which are the main existing metrics for evaluating black-box abstaining classifiers. As a two-dimensional metric, comparison on the basis of these is non-trivial. A common approach is to assume a fixed cost for each abstention (Chow, 1970), but this is not always satisfactory since determining how to weigh abstentions and errors against one another is a nontrivial question. Thus, in settings such as Example 1.1, the notion of counterfactual score becomes necessary. Importantly, selective scores are not representative of the counterfactual performance, except in the (unrealistic) case wherein predictions are missing completely at random (MCAR).[2] Figure 1 gives an overview of scenarios where different metrics may be appropriate to compare abstaining classifiers.

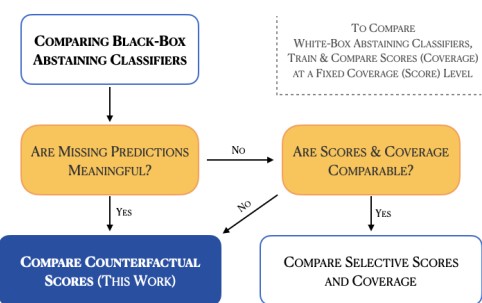

Figure 1: A schematic flowchart of comparing abstaining classifiers. In a black-box setting where the evaluator does not have access to the training algorithms or the resources to train them, the task can be viewed as a nontrivial missing data problem. This work proposes the counterfactual score as an evaluation metric.

The counterfactual score also offers practical benefits when comparing abstaining classifiers. Counterfactual scores are comparable even if abstaining classifiers are tuned to different abstention rates or selective scores. Moreover, compared to evaluation methods using the selective score-coverage curve (equivalent to re-training the classifier several times at different score/coverage levels), estimating the counterfactual score does not require re-training the classifier. Instead, we only need to estimate a pair of nuisance functions that can be learned using the observed predictions (nonabstentions) in the evaluation set. Let us further note that the setup is applicable generally to any form of prediction that can be scored, including regression and structured prediction. In this paper, we restrict our attention to classification for concreteness, as it is the most well-studied abstention framework.

**Summary of contributions**  We first formalize the problem of comparing abstaining classifiers as a missing data problem and introduce the counterfactual score as a novel metric for abstaining classifiers. Next, we discuss how the counterfactual score can be identified under the MAR and positivity conditions. Then, we develop efficient nonparametric estimators for the counterfactual scores and their differences, namely doubly robust confidence intervals (DR CI). Finally, we validate our approach in simulated and real-data experiments.

---

[2]MCAR means the missing observations are simply a uniformly random subset of all observations, independently of the input/output. In contrast, MAR means there can be systematic differences between the missing and observed values, but these can be explained by the input. Our method only requires MAR.

**Related work on abstaining classifiers** The *training* of abstaining classifiers has seen significant interest in the literature (e.g., Chow, 1970; Pietraszek, 2005; Bartlett and Wegkamp, 2008; Cortes et al., 2016; Geifman and El-Yaniv, 2017, 2019; Gangrade et al., 2021). We refer to Hendrickx et al. (2021); Zhang et al. (2023) for recent surveys. For *evaluation*, aside from using some combination of selective score and coverage, the most pertinent reference is the work of Condessa et al. (2017), who propose the metric of 'classifier quality' that is a somewhat inverse version of our counterfactual accuracy. This metric is the sum of the prediction accuracy when the classifier predicts, and prediction *inaccuracy* when it abstains, the idea being that if a classifier is not abstaining needlessly, then it must hold that the underlying predictions on points it abstains on are very poor. While this view is relevant to the training of abstention rules, it is at odds with black-box settings where the underlying predictions may still be executed even when the method abstains, motivating the counterfactual score. In terms of estimating the score, Condessa et al. (2017) do not discuss the black-box setting that requires estimating a counterfactual, as they assume full knowledge of the classifier's predictions.

**Related work on missing data, causal inference, and doubly robust estimation** Our main approach to the estimability of counterfactual scores is driven by a reduction to an inference problem under missing data (or censoring) (Rubin, 1976; Little and Rubin, 2019). A missing data problem can be viewed equivalently as a causal inference problem in an observational study (Rubin, 1976; Pearl, 2000; Shpitser et al., 2015; Ding and Li, 2018), and there exist well-established theories and methods for both identifying the counterfactual quantity of interest and efficiently estimating the identified target functional. For *identification*, unlike in some observational settings for estimating treatment effects, our setup straightforwardly satisfies the standard assumption of consistency (non-interference). The MAR assumption is satisfied as long as independent evaluation data is used, and the positivity assumption translates to abstentions being stochastic. We discuss various implications of these conditions in Sections 2.2 and 5. For *estimation*, efficient methods for estimating targets such as the average treatment effect (ATE) have long been studied under semiparametric and nonparametric settings. Doubly robust (DR) estimators, in particular, are known to achieve the asymptotic minimax lower bound on the mean squared error. For details, we refer the reader to Bickel et al. (1993); Robins et al. (1994); van der Vaart (2000, 2002); van der Laan and Robins (2003); Bang and Robins (2005); Tsiatis (2006); Chernozhukov et al. (2018); Kennedy (2022). Unlike the standard ATE estimation setup in observational studies, our setup contrasts two separate causal estimands, the counterfactual scores of the two competing classifiers, that operate under their distinct missingness mechanisms.

## 2   Definition and identification of the counterfactual score

We formulate the problem of evaluating and comparing abstaining classifiers under the missing data framework (Rubin, 1976). We follow the standard approach of defining the target parameter (Section 2.1), identifying it with observable quantities (Section 2.2), and estimating the identified parameter using data (Section 3). In each step, we first consider evaluating one abstaining classifier and then extend to comparing two abstaining classifiers. Below, $\mathcal{X}$ denotes the input space and $\mathcal{Y} = \{1, \ldots, C\}$ is the set of possible classes, while $\Delta^{C-1}$ denotes the $C$-dimensional probability simplex on $\mathcal{Y}$.

**Abstaining classifiers** We define an abstaining classifier as a pair of functions $(f, \pi)$, representing its *base classifier* $f : \mathcal{X} \to \Delta^{C-1}$ and *abstention mechanism* $\pi : \mathcal{X} \to [0, 1]$, respectively. Given a query $X$, the classifier first forms a preliminary (probabilistic) prediction $f(X)$. Then, potentially using the output $f(X)$, the classifier determines $\pi(X)$, i.e., the abstention probability. Using $\pi(X)$, the classifier then makes the binary abstention decision $R \mid \pi(X) \sim \mathsf{Ber}(\pi(X))$, so that if $R = 1$ ("rejection"), the classifier abstains on the query, and if $R = 0$, it reveals its prediction $f(X)$. In some cases, we will explicitly define the source of randomness $\xi$ (independent of the data) in deciding $R$, such that $R = \mathsf{r}(\pi(X), \xi)$ for a deterministic function $\mathsf{r}$.[3] Neither $f$ nor $\pi$ is assumed to be known to the evaluator, modeling the black-box access typically available to practitioners.

**Scoring rules (Higher scores are better.)** We measure the quality of a prediction $f(x)$ for a label $y$ via a positively oriented *scoring rule* $\mathsf{s} : \Delta^{C-1} \times \mathcal{Y} \to \mathbb{R}$. One simple scoring rule is classification accuracy, i.e., $\mathsf{s}(f(x), y) = \mathbf{1}\left(\mathrm{argmax}_{c \in \mathcal{Y}} f(x)_c = y\right)$, but a plethora of scores exist in the literature, such as the Brier (1950) score: $\mathsf{s}(f(x), y) = 1 - \sum_{c \in \mathcal{Y}} (f(x)_c - \mathbf{1}(y = c))^2$.

---

[3]Specifically, let $\xi \sim \mathsf{Unif}[0, 1]$ and $R = \mathbf{1}(\xi \leq \pi(X))$. Then, $R$ is a function of only $\pi(X)$ and $\xi$.

**The evaluation setup** For each labeled data point $(X, Y)$ in an evaluation set, we observe the abstention decision $R = r(\pi(X), \xi)$ for some independent source of randomness $\xi$ used by the abstaining classifier. Then, its prediction $f(X)$ is observed by the evaluator if and only if $R = 0$. Let $S := s(f(X), Y)$ denote the score of the prediction $f$ on the query $X$, irrespective of $R$. Because $S$ is not observable when $R = 1$, we refer to $S$ as the *potential score* that *would have been seen* had the classifier not abstained. (See Appendix A.2 for equivalent formulations that explicitly invoke Rubin (1974)'s potential outcomes model.)

Since our evaluation is based only on the score $S$, we can suppress the role of $Y$ and assume that $S$ is observed directly when $R = 0$. Similarly, we can suppress the role of $\xi$, which is independent of the data. We let $\mathbb{P}$ denote the law of $Z := (X, R, S)$.

Table 1 summarizes the problem formulations and the proposed approaches in our paper. The target definitions and the identification conditions are discussed in the following subsections; the estimator and its optimality are discussed in Section 3.

Table 1: A summary of problem formulations and proposed approaches for evaluation and comparison of abstaining classifiers. Our approaches avoid parametric assumptions and allow for black-box classifiers.

|  | **Evaluation** | **Comparison** |
| --- | --- | --- |
| Classifier(s) | $(f, \pi)$ | $(f^A, \pi^A)$ & $(f^B, \pi^B)$ |
| Target | $\psi = \mathbb{E}[S]$ | $\Delta^{AB} = \mathbb{E}[S^A - S^B]$ |
| Identification | MAR & positivity | |
| Estimation | Doubly robust CI | |
| Optimality | Nonparametrically efficient | |

## 2.1 The counterfactual score

We propose to assess an abstaining classifier $(f, \pi)$ with its *(expected) counterfactual score*:

$$\psi := \mathbb{E}[S], \tag{2.1}$$

where the expectation is taken w.r.t. $\mathbb{P}$. In words, $\psi$ refers to the expected score of the abstaining classifier had it not been given the option to abstain. The counterfactual score captures the performance of an abstaining classifier via the score of its base classifier, making it suitable for cases where the evaluator is interested in the predictions without using an abstention mechanism.

Note that $\psi$ does *not* in general equal the *selective score*, i.e., $\mathbb{E}[S \mid R = 0]$. For example, when a classifier abstains from making predictions on its "weak points," i.e., inputs on which the classifier performs poorly, the counterfactual score will be lower than the selective score. Also see Appendix A.3 for a direct comparison with Condessa et al. (2017)'s score, which *rewards* having bad hidden predictions, as opposed to (2.1). Our general framework allows estimating either score, although we focus on (2.1) based on our motivating examples (Example 1.1 and Appendix A.1).

**Comparison** Counterfactual scores may also be used to compare two abstaining classifiers, $(f^A, \pi^A)$ and $(f^B, \pi^B)$, in the form of their *counterfactual score difference*: $\Delta^{AB} := \psi^A - \psi^B = \mathbb{E}[S^A - S^B]$. Here, the expectation is now taken over the joint law of $Z^{AB} := (X, R^A, S^A, R^B, S^B)$.

## 2.2 Identifiability of the counterfactual score

Having defined the target parameters $\psi$ and $\Delta^{AB}$, we now discuss the assumptions under which these quantities become identifiable using only the observed random variables. In other words, these assumptions establish when the counterfactual quantity equals a statistical quantity. As in standard settings of counterfactual inference under missing data, the identifiability of counterfactual scores in this setting depends on two standard conditions: (i) the missing at random condition and (ii) positivity.

The *missing at random (MAR)* condition, also known as the *ignorability* or *no unmeasured confounding* condition, requires that the score $S$ is conditionally independent of the abstention decision $R$ given $X$, meaning that there are no unobserved confounders $U$ that affect both the abstention decision $R$ as well as the score $S$. Note that $S$ is the *potential* score of what the classifier would get had it not abstained — it is only observed when $R = 0$. We formally state the MAR condition as follows:

**Assumption 2.1** (Scores are missing at random). $S \perp\!\!\!\perp R \mid X$.

In standard ML evaluation scenarios, where the evaluation set is independent of the training set for the classifier, Assumption 2.1 is always met. We formalize this sufficient condition for MAR in the following proposition. Let $\mathcal{D}_{\text{train}}$ denote the collection of any training data used to learn the abstaining classifier $(f, \pi)$ and, as before, $(X, Y)$ denote an (i.i.d.) data point in the evaluation set.

**Proposition 2.2** (Independent evaluation data ensures MAR). *If $(X, Y) \perp\!\!\!\perp \mathcal{D}_{\text{train}}$, then $S \perp\!\!\!\perp R \mid X$.*

This result is intuitive: given an independent test input $X$, the score $S = \mathsf{s}(f(X), Y)$ is a deterministic function of the test label $Y$, and the abstention decision $R$ of a classifier cannot depend on $Y$ simply because the classifier has no access to it. A short proof is given in Appendix B.1 for completeness. In Appendix C, we also include causal graphs that visually illustrate how the MAR condition is met.

If the evaluation data is not independent of $\mathcal{D}_{\text{train}}$, then the classifier already has information about the data on which it is tested, so generally speaking, no evaluation score will not accurately reflect its generalization performance. Although the independence between the training and evaluation data is expected in standard ML applications, it may not be guaranteed when, e.g., using a publicly available dataset that is used during the training of the classifier. These issues can be prevented by ensuring that the evaluation set is held out (e.g., a hospital can use its own patient data to evaluate APIs).

The second condition, the *positivity* condition, is more substantial in our setting:

**Assumption 2.3** (Positivity). There exists $\epsilon > 0$ such that $\pi(X) = \mathbb{P}(R = 1 \mid X) \leq 1 - \epsilon$.

Assumption 2.3 says that, for each input $X$, there has to be at least a small probability that the classifier will *not* abstain ($R = 0$). Indeed, if the classifier deterministically abstains from making predictions on a specific input that has nonzero marginal density, then we have no hope of estimating an expectation over all possible values that $X$ can take. When it comes to evaluating abstaining classifiers on safety-critical applications, we argue that this condition may need to be enforced at a policy level — we elaborate on this point in Section 5 and Appendix A.5. In practice, the exact value of $\epsilon$ is problem-dependent, and in Appendix E.4, we include additional experiments illustrating how our methods retain validity as long as the abstention rate is capped at $1 - \epsilon$ for some $\epsilon > 0$.

Another justification for the positivity condition is that stochastically abstaining classifiers can achieve better performances than their deterministic counterparts. Kalai and Kanade (2021) illustrate how stochastic abstentions can improve the out-of-distribution (OOD) performance w.r.t. the Chow (1970) score (i.e., $\mathrm{selective\ score} + \alpha \cdot \mathrm{coverage}$). Schreuder and Chzhen (2021) also introduce randomness in their abstaining classifiers, which leverage abstentions as a means to improve their accuracy while satisfying a fairness constraint. The role of random abstentions in these examples mirrors the role of randomization in the fairness literature (Barocas et al., 2019), where the optimal randomized fair predictors are known to outperform their deterministic counterparts (Agarwal and Deshpande, 2022; Grgić-Hlača et al., 2017). Given the effectiveness of randomized classifiers for fairness, it would not be surprising if a fair abstaining classifier was randomized (in its decisions and abstentions).

With MAR and positivity in hand, we can show that the counterfactual score is indeed identifiable. Define $\mu_0(x) := \mathbb{E}[S \mid R = 0, X = x]$ as the regression function for the score under $R = 0$.

**Proposition 2.4** (Identification). *Under Assumptions 2.1 and 2.3, $\psi$ is identified as $\mathbb{E}[\mu_0(X)]$.*

The proof, included in Appendix B.2, follows a standard argument in causal inference. The identification of the target parameter $\psi$ using $\mu_0$ implies that we can estimate $\psi$, the expectation of a potential outcome, using only the observed inputs and scores. Specifically, the task of estimating $\psi$ consistently reduces to the problem of estimating the regression function $\mu_0$, which only involves predictions that the classifier did not abstain from making. We note that, as in standard causal inference, the task of identification, which concerns *what* to estimate, is largely orthogonal to the task of estimation, which concerns *how* to estimate the quantity. We discuss the latter problem in Section 3.

**Comparison**  For the comparison task, given $\Delta^{\mathsf{AB}} = \psi^{\mathsf{A}} - \psi^{\mathsf{B}}$, it immediately follows that if the MAR and positivity assumptions hold for each of $(X, R^{\mathsf{A}}, S^{\mathsf{A}})$ and $(X, R^{\mathsf{B}}, S^{\mathsf{B}})$, then $\Delta^{\mathsf{AB}}$ is also identified as $\Delta^{\mathsf{AB}} = \mathbb{E}[\mu_0^{\mathsf{A}}(X) - \mu_0^{\mathsf{B}}(X)]$, where $\mu_0^{\bullet}(x) := \mathbb{E}[S^{\bullet} \mid R^{\bullet} = 0, X = x]$ for $\bullet \in \{\mathsf{A}, \mathsf{B}\}$. In words, if (i) the evaluation data is independent of the training data for each classifier (Proposition 2.2) and (ii) each classifier has at least a small chance of not abstaining on each input (Assumption 2.3), then the counterfactual score difference, i.e., $\Delta^{\mathsf{AB}}$, is identified as the expected difference in the expected scores over inputs conditional on non-abstentions, i.e., $\mathbb{E}[\mu_0^{\mathsf{A}}(X) - \mu_0^{\mathsf{B}}(X)]$.

## 3 Nonparametric and doubly robust estimation of the counterfactual score

Having identified the counterfactual scores, we now focus on the problem of consistently estimating them. We estimate these quantities without resorting to parametric assumptions about the underlying

black-box abstention mechanisms. Instead, we reduce the problem to that of functional estimation and leverage techniques from nonparametric statistics. See Kennedy (2022) for a recent review.

## 3.1 Estimating the counterfactual score

**Task** Let $\{(X_i, R_i, S_i)\}_{i=1}^n \sim \mathbb{P}$ denote an i.i.d. evaluation set of size $n$. As before, we assume that we are given access to the censored version of this sample, i.e., that we observe $S_i$ if and only if $R_i = 0$. Using the observables, we seek to form an estimate $\hat{\psi}$ of the counterfactual score $\psi = \mathbb{E}[S]$.

**Doubly robust estimation** Under identification (Proposition 2.4), we can estimate $\psi$ by estimating the regression function $\mu_0(X)$ on the data $\{(X_i, S_i) : R_i = 0\}$. However, the naïve "plug-in" estimate suffers from an inflated bias due to the structure present in the abstention patterns. (See Appendix A.4 for details.) We instead develop a doubly robust (DR) estimator (Robins et al., 1994; Bang and Robins, 2005), which is known to consistently estimate $\psi$ at the optimal *nonparametric efficiency rates*, meaning that no other estimator based on the $n$ observations can asymptotically achieve a smaller mean squared error (van der Vaart, 2002). The derivation below is relatively standard, explaining our brevity.

Formally, the DR estimator is defined using the (uncentered) *efficient influence function (EIF)* for the identified target functional $\psi(\mathbb{P}) = \mathbb{E}_{\mathbb{P}}[\mu_0(X)]$: $\mathsf{IF}(x, r, s) := \mu_0(x) + \frac{1-r}{1-\pi(x)}(s - \mu_0(x))$ $(0/0 := 0)$. Here, $\pi$ and $\mu_0$ are the "nuisance" functions, representing the abstention mechanism and the score function under $R = 0$, respectively. The EIF can be computed as long as $s$ is available when $r = 0$. An intuition for the EIF is that it is the first-order "distributional Taylor approximation" (Fisher and Kennedy, 2021) of the target functional, such that its bias is second-order.

Given that $\pi$ and $\mu_0$ are unknown, we define an estimate of the EIF, denoted as $\hat{\mathsf{IF}}$, by plugging in estimates $\hat{\pi}$ for $\pi$ and $\hat{\mu}_0$ for $\mu_0$. Then, the DR estimator is simply the empirical mean of the EIF:

$$\hat{\psi}_{\mathsf{dr}} = \frac{1}{n}\sum_{i=1}^n \hat{\mathsf{IF}}(X_i, R_i, S_i) = \frac{1}{n}\sum_{i=1}^n \left[\hat{\mu}_0(X_i) + \frac{1-R_i}{1-\hat{\pi}(X_i)}(S_i - \hat{\mu}_0(X_i))\right]. \qquad (3.1)$$

This estimator is well-defined because $S_i$ is available precisely when $R_i = 0$. Note that the first term is the (biased) plug-in estimator, and the second term represents the first-order correction term, which involves inverse probability weighting (IPW) (Horvitz and Thompson, 1952; Rosenbaum, 1995). In our experiments, we show how the DR estimator improves upon both the plug-in estimator, in terms of the bias, and the IPW-based estimator, which we recap in Section A.4, in terms of the variance.

The "double robustness" of $\hat{\psi}_{\mathsf{dr}}$ translates to the following useful property: $\hat{\psi}_{\mathsf{dr}}$ retains the parametric rate of convergence, $O_{\mathbb{P}}(1/\sqrt{n})$, even when the estimators $\hat{\mu}_0$ and $\hat{\pi}$ themselves converge at slower rates. This allows us to use nonparametric function estimators to estimate $\mu_0$ and $\pi$, such as stacking ensembles (Breiman, 1996) like the super learner (van der Laan et al., 2007) and regularized estimators like the Lasso (Tibshirani, 1996; Belloni et al., 2014). Even for nonparametric models whose rates of convergence are not fully understood, such as random forests (Breiman, 2001) and deep neural networks (LeCun et al., 2015), we can empirically demonstrate valid coverage and efficiency (Section 4).

In practice, the nuisance functions can be estimated via *cross-fitting* (Robins et al., 2008; Zheng and van der Laan, 2011; Chernozhukov et al., 2018), which is a $K$-fold generalization of sample splitting. First, randomly split the data into $K$ folds; then, fit $\hat{\pi}$ and $\hat{\mu}_0$ on $K-1$ folds and use them to estimate the EIF on the remaining "evaluation" fold; repeat the process $K$ times with each fold being the evaluation fold; finally, average the EIFs across all data points. The key benefit of using cross-fitting is to avoid any complexity restrictions on individual nuisance functions without sacrificing sample efficiency. In the following, we let $\hat{\psi}_{\mathsf{dr}}$ be the estimator (3.1) obtained via cross-fitting.

Now we are ready to present our first result, which states the asymptotic validity and efficiency of the DR estimator for $\psi$ under identification and the DR condition.

**Theorem 3.1** (DR estimation of the counterfactual score for an abstaining classifier). *Suppose that Assumptions 2.1 and 2.3 hold. Also, suppose that*

$$\|\hat{\pi} - \pi\|_{L_2(\mathbb{P})} \|\hat{\mu}_0 - \mu_0\|_{L_2(\mathbb{P})} = o_{\mathbb{P}}(1/\sqrt{n}) \qquad (3.2)$$

*and that* $\|\hat{\mathsf{IF}} - \mathsf{IF}\|_{L_2(\mathbb{P})} = o_{\mathbb{P}}(1)$. *Then,*

$$\sqrt{n}\left(\hat{\psi}_{\mathsf{dr}} - \psi\right) \rightsquigarrow \mathcal{N}\left(0, \mathsf{Var}_{\mathbb{P}}\left[\mathsf{IF}\right]\right),$$

*where* $\mathsf{Var}_{\mathbb{P}}\left[\mathsf{IF}\right]$ *matches the nonparametric efficiency bound.*

The proof adapts standard arguments in mathematical statistics, as found in, e.g., van der Vaart (2002); Kennedy (2022), to the abstaining classifier evaluation setup. We include a proof sketch in Appendix B.3. Theorem 3.1 tells us that, under the identification and the DR condition (3.2), we can construct a closed-form asymptotic confidence interval (CI) at level $\alpha \in (0,1)$ as follows:

$$C_{n,\alpha} = \left(\hat{\psi}_{\mathsf{dr}} \pm z_{\alpha/2}\sqrt{n^{-1}\mathsf{Var}_{\hat{\mathbb{P}}_n}[\hat{\mathsf{IF}}]}\right), \tag{3.3}$$

where $z_{\alpha/2} = \Phi(1 - \frac{\alpha}{2})$ is the $(1 - \frac{\alpha}{2})$-quantile of a standard normal (e.g., 1.96 for $\alpha = 0.05$). In Appendix D, we describe an asymptotic confidence sequence (AsympCS; Waudby-Smith et al., 2021), which is a CI that is further valid under continuous monitoring (e.g., as more data is collected).

## 3.2 Estimating counterfactual score differences

**Task**   Next, we return to the problem of comparing two abstaining classifiers, $(f^{\mathsf{A}}, \pi^{\mathsf{A}})$ and $(f^{\mathsf{B}}, \pi^{\mathsf{B}})$, that each makes a decision to make a prediction on each input $X_i$ or abstain from doing so. That is, $R_i^{\bullet} \mid \pi^{\bullet}(X_i) \sim \mathsf{Ber}(\pi^{\bullet}(X_i))$, and we observe $S_i^{\bullet} = \mathsf{s}(f^{\bullet}(X_i), Y_i)$ if and only if $R_i^{\bullet} = 0$, for $\bullet \in \{\mathsf{A}, \mathsf{B}\}$. Recall that the target here is the score difference $\Delta^{\mathsf{AB}} = \psi^{\mathsf{A}} - \psi^{\mathsf{B}} = \mathbb{E}\left[S^{\mathsf{A}} - S^{\mathsf{B}}\right]$.

**Doubly robust difference estimation**   If the parameters $\psi^{\mathsf{A}}$ and $\psi^{\mathsf{B}}$ are each identified according to Proposition 2.4, then we can estimate $\Delta^{\mathsf{AB}}$ as $\hat{\Delta}^{\mathsf{AB}} = \hat{\psi}^{\mathsf{A}} - \hat{\psi}^{\mathsf{B}}$, for individual estimates $\hat{\psi}^{\mathsf{A}}$ and $\hat{\psi}^{\mathsf{B}}$. The resulting EIF is simply the difference in the EIF for A and B: $\mathsf{IF}^{\mathsf{AB}}(x, r^{\mathsf{A}}, r^{\mathsf{B}}, s^{\mathsf{A}}, s^{\mathsf{B}}) = \mathsf{IF}^{\mathsf{A}}(x, r^{\mathsf{A}}, s^{\mathsf{A}}) - \mathsf{IF}^{\mathsf{B}}(x, r^{\mathsf{B}}, s^{\mathsf{B}})$, where $\mathsf{IF}^{\mathsf{A}}$ and $\mathsf{IF}^{\mathsf{B}}$ denote the EIF of the respective classifier. Thus, we arrive at an analogous theorem that involves estimating the nuisance functions of each abstaining classifier and utilizing $\mathsf{IF}^{\mathsf{AB}}$ to obtain the limiting distribution of $\hat{\Delta}_{\mathsf{dr}}^{\mathsf{AB}} = \hat{\psi}_{\mathsf{dr}}^{\mathsf{A}} - \hat{\psi}_{\mathsf{dr}}^{\mathsf{B}}$.

**Theorem 3.2** (DR estimation of the counterfactual score difference)**.** *Suppose that Assumptions 2.1 and 2.3 hold for both $(X_i, R_i^{\mathsf{A}}, S_i^{\mathsf{A}})$ and $(X_i, R_i^{\mathsf{B}}, S_i^{\mathsf{B}})$. Also, suppose that*

$$\|\hat{\pi}^{\mathsf{A}} - \pi^{\mathsf{A}}\|_{L_2(\mathbb{P})}\|\hat{\mu}_0^{\mathsf{A}} - \mu_0^{\mathsf{A}}\|_{L_2(\mathbb{P})} + \|\hat{\pi}^{\mathsf{B}} - \pi^{\mathsf{B}}\|_{L_2(\mathbb{P})}\|\hat{\mu}_0^{\mathsf{B}} - \mu_0^{\mathsf{B}}\|_{L_2(\mathbb{P})} = o_{\mathbb{P}}(1/\sqrt{n}) \tag{3.4}$$

*and that* $\|\hat{\mathsf{IF}}^{\mathsf{AB}} - \mathsf{IF}^{\mathsf{AB}}\|_{L_2(\mathbb{P})} = o_{\mathbb{P}}(1)$. *Then,*

$$\sqrt{n}\left(\hat{\Delta}_{\mathsf{dr}}^{\mathsf{AB}} - \Delta^{\mathsf{AB}}\right) \rightsquigarrow \mathcal{N}\left(0, \mathsf{Var}_{\mathbb{P}}\left[\mathsf{IF}^{\mathsf{AB}}\right]\right),$$

*where* $\mathsf{Var}_{\mathbb{P}}[\mathsf{IF}^{\mathsf{AB}}]$ *matches the nonparametric efficiency bound.*

A proof is given in Appendix B.4. As with evaluation, Theorem 3.2 yields a closed-form asymptotic CI of the form (3.3) using the analogous estimate of EIF under MAR, positivity, and DR (3.4). Inverting this CI further yields a hypothesis test for $H_0 : \psi^{\mathsf{A}} = \psi^{\mathsf{B}}$ vs. $H_1 : \psi^{\mathsf{A}} \neq \psi^{\mathsf{B}}$.

# 4   Experiments

The main goals of our experiments are to empirically check the validity and power of our estimation methods and to illustrate how our methods can be used in practice. First, in Section 4.1, we present results on simulated data to examine the validity of our proposed inference methods (CIs and hypothesis tests). Then, in Section 4.2, we study three scenarios on the CIFAR-100 dataset that illustrate the practical use of our approach to real data settings. All code for the experiments is publicly available online at `https://github.com/yjchoe/ComparingAbstainingClassifiers`.

## 4.1   Simulated experiments: Abstentions near the decision boundary

**Setup (MAR but not MCAR)**   We first consider comparing two abstaining classifiers according to their accuracy scores, on a simulated binary classification dataset with 2-dimensional inputs.

Given $n = 2,000$ i.i.d. inputs $\{X_i\}_{i=1}^n \sim \text{Unif}([0,1]^2)$, each label $Y_i$ is decided using a linear boundary, $f_*(x_1, x_2) = \mathbf{1}(x_1 + x_2 \geq 1)$, along with a 15% i.i.d. label noise. Importantly, each classifier abstains near its decision boundary, such that its predictions and scores are *MAR but not MCAR* (because abstentions depend on the inputs). As a result, while the counterfactual score of A ($\psi^A = 0.86$) is much higher than B ($\psi^B = 0.74$), their selective scores are more similar (Sel$^A = 0.86$, Sel$^B = 0.81$) and coverage is lower for A (Cov$^A = 0.55$) than for B (Cov$^B = 0.62$). Another point to note here is that, even though both the outcome model and classifier A are linear, both the abstention mechanism[4] $\pi^A$ and the selective score function[5] $\mu_0^A$ are *non*linear functions of the inputs (similarly for $\pi^B$ and $\mu_0^B$). More generally, if a classifier abstains near its decision boundary, then both $\pi$ and $\mu_0$ could be at least as complex as the base classifier $f$ itself. Further details of the setup, including a plot of the data, predictions, and abstentions, are provided in Appendix E.1.

**Miscoverage rates and widths**  As our first experiment, we compare the miscoverage rates and widths of the 95% DR CIs (Theorem 3.2) against two baseline estimators: the plug-in and the IPW (Rosenbaum, 1995) estimators (Appendix A.4). For each method, the miscoverage rate of the CI $C_n$ is approximated via $\mathbb{P}(\Delta^{AB} \notin C_n) \approx m^{-1} \sum_{j=1}^m \mathbf{1}(\Delta^{AB} \notin C_n^{(j)})$, where $m$ is the number of simulations over repeatedly sampled data. If the CI is valid, then this rate should approximately be $0.05$. The miscoverage rate and the width of a CI, respectively, capture its bias and variance components. For the nuisance functions, we try linear predictors (L2-regularized linear/logistic regression for $\hat{\mu}_0/\hat{\pi}$), random forests, and super learners with $k$-NN, kernel SVM, and random forests.

We present our results in Table 2. First, using the random forest or the super learner,

Table 2: Miscoverage rates (and widths) of 95% CIs using three estimation approaches and three nuisance function ($\pi$ and $\mu_0$) estimators in a simulated experiment. Mean and standard error computed over $m = 1,000$ runs are shown; those within 2 standard errors of the intended level ($0.05$) are boldfaced. The sample size is $n = 2,000$ in each run. The mean widths of CIs are shown in parentheses. DR estimation with either a random forest or a super learner achieves control over the miscoverage rate, and the DR-based CI is twice as tight as the IPW-based CI in terms of their width.

| $\hat{\mu}_0$ & $\hat{\pi}$ | Plug-in | IPW | DR |
|---|---|---|---|
| Linear | $1.00 \pm 0.00$ $(0.00)$ | $0.76 \pm 0.01$ $(0.09)$ | $1.00 \pm 0.00$ $(0.04)$ |
| Random forest | $0.64 \pm 0.02$ $(0.02)$ | $0.14 \pm 0.01$ $(0.13)$ | $\mathbf{0.05 \pm 0.01}$ $\mathbf{(0.07)}$ |
| Super learner | $0.91 \pm 0.01$ $(0.01)$ | $\mathbf{0.03 \pm 0.01}$ $\mathbf{(0.12)}$ | $\mathbf{0.05 \pm 0.01}$ $\mathbf{(0.06)}$ |

the DR CIs consistently achieve the intended coverage level of $0.95$, over $m = 1,000$ repeated simulations (standard error $0.01$). This validates the asymptotic normality result of (3.2). Note that the CI with linear estimators does not achieve the intended coverage level: this is expected as neither $\hat{\pi}$ nor $\hat{\mu}_0$ can consistently estimate the nonlinear functions $\pi$ or $\mu_0$, violating the DR condition (3.4).

Second, when considering both the miscoverage rate and CI width, the DR estimator outperforms both the plug-in and IPW estimators. The plug-in estimator, despite having a small CI width, has a very high miscoverage rate ($0.91$ with the super learner), meaning that it is biased even when flexible nuisance learners are used. On the other hand, the IPW estimator has double the width of the DR estimator ($0.12$ to $0.06$, with the super learner), meaning that it is not as efficient. Also, while the IPW estimator achieves the desired coverage level with the super learner ($0.03$), it fails with the random forest ($0.14$), which tends to make overconfident predictions of the abstention pattern and biases the resulting estimate. In contrast, the DR estimator retains its intended coverage level of $0.05$ with the random forest, suggesting that it is amenable to overconfident nuisance learners.

**Power analysis**  We conduct a power analysis of the statistical test for $H_0 : \Delta^{AB} = 0$ vs. $H_1 : \Delta^{AB} \neq 0$ by inverting the DR CI. The results confirm that the power reaches 1 as either the sample size ($n$) or the absolute difference ($|\Delta^{AB}|$) increases. This experiment is included in Appendix E.2.

---

[4]The abstention mechanism $\pi(x) = \mathbb{P}(R = 1 \mid X = x)$ here separates the region below *and* above the decision boundary from the region near the boundary. Thus $\pi$ is nonlinear even when the boundary is linear.

[5]Given any input $X$ for which the classifier did not abstain ($R = 0$) and its output $Y$, the score $S = \mathsf{s}(f(X), Y)$ is nonlinear if either $\mathsf{s}(\cdot, y)$ or $f$ is nonlinear. Thus, even for linear $f$, nonlinear scores like the Brier score automatically make the selective score function $\mu_0(x) = \mathbb{E}[S \mid R = 0, X = x]$ nonlinear.

## 4.2 Comparing abstaining classifiers on CIFAR-100

To illustrate a real data use case, we compare abstaining classifiers on the CIFAR-100 image classification dataset (Krizhevsky, 2009). Observe that abstaining classifiers can behave differently not just when their base classifiers are different but also when their abstention mechanisms are different. In fact, two abstaining classifiers can have a similarly effective base classifier but substantially different abstention mechanisms (e.g., one more confident than the other). In such a case, the counterfactual score difference between the two classifiers is zero, but their selective scores and coverages are different. We examine such scenarios by comparing image classifiers that use the same pre-trained representation model but have different output layers and abstention mechanisms.

We start with the 512-dimensional final-layer representations of a VGG-16 convolutional neural network (Simonyan and Zisserman, 2015), pre-trained[6] on the CIFAR-100 training set, and compare different output layers and abstention mechanisms on the validation set. Generally, in a real data setup, we cannot verify whether a statistical test or a CI is correct; yet, in this experiment, we can still access to the base model of each abstaining classifier. This means that (a) if we compare abstaining classifiers that share the base classifier but differ in their abstention patterns, then we actually know that their counterfactual scores are exactly the same ($\Delta^{AB} = 0$); (b) if we compare abstaining classifiers with different base classifiers, then we can compute their counterfactual scores accurately up to an i.i.d. sampling error. This estimate is denoted by $\bar{\Delta}^{AB} := n^{-1} \sum_{i=1}^{n} [\mathsf{s}(f^A(X_i), Y_i) - \mathsf{s}(f^B(X_i), Y_i)]$.

For all comparisons, we use the DR estimator, where the nuisance functions $\hat{\pi}$ and $\hat{\mu}_0$ for both classifiers are each an L2-regularized linear layer learned on top of the pre-trained VGG-16 features. The use of pre-trained representations for learning the nuisance functions is motivated by Shi et al. (2019), who demonstrated the effectiveness of the approach in causal inference contexts. We also use the Brier score in this experiment. We defer other experiment details to Appendix E.3.

In scenario I, we compare two abstaining classifiers that use the same softmax output layer but use a different threshold for abstentions. Specifically, both classifiers use the softmax response (SR) thresholding (Geifman and El-Yaniv, 2017), i.e., abstain if $\max_{c \in \mathcal{Y}} f(X)_c < \tau$ for a threshold $\tau > 0$, but A uses a more conservative threshold ($\tau = 0.8$) than B ($\tau = 0.5$). As a result, while their counterfactual scores are identical ($\Delta^{AB} = 0$), A has a higher selective score ($+0.06$) and a lower coverage ($-0.20$) than B. This is also a deterministic abstention mechanism, potentially challenging the premises of our setup. As shown in Table 3, we see that the 95% DR CI is $(-0.005, 0.018)$ ($n = 5,000$), confirming that there is no difference in counterfactual scores.[7]

Table 3: The 95% DR CIs and their corresponding hypothesis tests for $H_0 : \Delta^{AB} = 0$ at significance level $\alpha = 0.05$, for three different comparison scenarios on (half of) the CIFAR-100 test set ($n = 5,000$). The three scenarios compare different abstention mechanisms or predictors, as detailed in text; all comparisons use the Brier score. $\bar{\Delta}^{AB}$ is the empirical counterfactual score difference without any abstentions. The result of each statistical test agrees with whether $\bar{\Delta}^{AB}$ is 0.

| Scenarios | $\bar{\Delta}^{AB}$ | 95% DR CI | Reject $H_0$? |
|---|---|---|---|
| I | 0.000 | (-0.005, 0.018) | No |
| II | 0.000 | (-0.014, 0.008) | No |
| III | −0.029 | (-0.051, -0.028) | Yes |

Scenario II is similar to scenario I, except that the abstention mechanisms are now stochastic: A uses one minus the SR as the probability of making a prediction, i.e., $\pi^A(x) = 1 - \max_{c \in [C]} f(X)_c$, while B uses one minus the Gini impurity as the probability of abstention, i.e., $\pi^B(x) = 1 - \sum_{c=1}^{C} f(X)_c^2$, both clipped to $(0.2, 0.8)$. The Gini impurity is an alternative measure of confidence to SR that is quadratic in the probabilities, instead of piecewise linear, and thus the Gini-based abstaining classifier (B) is more cautious than the SR-based one (A), particularly on uncertain predictions. In our setup, A achieves a higher coverage than B, while B achieves a higher selective score than A. The 95% DR CI is $(-0.014, 0.008)$, confirming that there is once again no difference in counterfactual scores. Scenarios I and II both correspond to case (a).

In scenario III, we now examine a case where there *is* a difference in counterfactual scores between the two abstaining classifiers (case (b)). Specifically, we compare the pre-trained VGG-16 model's

---

[6]Reproduced version, accessed from `https://github.com/chenyaofo/pytorch-cifar-models`.

[7]The reason why the DR CI correctly estimates the true $\Delta^{AB} = 0$, despite the fact that positivity is violated in this case, is because the two classifiers happen to abstain on similar examples (B abstains whenever A does) *and* their scores on their abstentions happen to be relatively similar (0.604 for A; 0.576 for B).

output layers (512-512-100) with the single softmax output layer that we considered in earlier scenarios. It turns out that the original model's multi-layer output model achieves a worse Brier score (0.758) than one with a single output layer (0.787), likely because the probability predictions of the multi-layer model are too confident (miscalibrated). When using the same abstention mechanism (stochastic abstentions using SR, as in II), the overconfident original model correspondingly achieves a higher coverage (and worse selective Brier score) than the single-output-layer model. The Monte Carlo estimate of the true counterfactual score difference is given by $\hat{\Delta}^{AB} = -0.029$, and the 95% DR CI falls entirely negative with $(-0.051, -0.028)$, rejecting the null of $\Delta^{AB} = 0$ at $\alpha = 0.05$.

## 5 Discussion

This paper lays the groundwork for addressing the challenging problem of counterfactually comparing black-box abstaining classifiers. Our solution casts the problem in the missing data framework, in which we treat abstentions as MAR predictions of the classifier(s), and this allows us to leverage nonparametrically efficient tools from causal inference. We close with discussions of two topics.

**Addressing violations of the identification conditions** At the conceptual level, the biggest challenge arises from the positivity condition, which requires the classifiers to deploy a non-deterministic abstention mechanism. As mentioned in Section 2.2, the counterfactual score is unidentifiable without this assumption. We argue that this issue calls for a policy-level treatment, especially in auditing scenarios, where the evaluators may require vendors to supply a classifier that can abstain but has at least an $\epsilon > 0$ chance of nonabstention. The level $\epsilon$ can be mutually agreed upon by both parties. Such a policy achieves a middle ground in which the vendors are not required to fully reveal their proprietary classifiers. In Appendix A.5, we discuss this policy-level treatment in greater detail.

An alternative is to explore existing techniques that aim to address positivity violations directly (Petersen et al., 2012; Ju et al., 2019; Léger et al., 2022). Due to the unidentifiability result, these techniques have their own limitations. For example, we may consider applying sample trimming (Crump et al., 2009) to make valid inferences on a subpopulation, but the conclusions would not hold for the entire input domain (even when relevant). Nevertheless, it remains a meaningful future work to adapt modern tools in causal inference for diagnosing and addressing positivity violations (e.g., Lei et al., 2021; Kennedy, 2019). In the case of addressing MAR violations, if we suspect the evaluation data is "compromised" and used in training, we can adapt sensitivity analysis methods that measure how much of the evaluation data is contaminated by the training data (Bonvini and Kennedy, 2022).

**Connections to learning-to-defer scenarios and cascading classifiers** There are variants of abstaining classifiers in ML for which we can utilize the counterfactual score analogously. First, in the learning-to-defer setting (Madras et al., 2018), where the algorithm is allowed to defer its decision to an expert that gives their own decision, the counterfactual score naturally corresponds to the expected score of the overall system had the classifier not deferred at all. The score is thus an evaluation metric primarily for the classifier, and it is independent of the expert's predictions, even when the classifier is adaptive to the expert's tendencies. In the case where the goal is to assess the *joint* performance of the algorithm and the expert, then a variant of Condessa et al. (2017)'s score can be useful; in Appendix A.6, we discuss how the DR estimator can be utilized for estimating this variant, even when the expert's score is random.

Another example would be cascading or multi-stage classifiers (Alpaydin and Kaynak, 1998; Viola and Jones, 2001), which are classifiers that short-circuit easy predictions by using simple models in their early stages to save computation. The counterfactual framework can assess the performance of cascading classifiers by treating each component as an abstaining classifier. For example, a two-stage cascading classifier, equipped with a small model $f_0$, a large model $f_1$, and a deferral mechanism $\pi$, can be viewed as a pair of abstaining classifiers with tied abstention mechanisms, specifically $(f_0, \pi)$ and $(f_1, \bar{\pi})$ where $\bar{\pi} = 1 - \pi$. Then, the corresponding counterfactual scores capture the performance of the cascading classifier had it been using only that (small or large) model. We can further extend this to general multi-stage classifiers if we are interested in estimating the score of a classifier in each stage individually. The two-stage setup is also relevant to the learning-to-defer setup, where the large model is an imperfect expert on the task. In such a setup, we can devise a metric that combines (i) the selective score of the small model and (ii) the relative improvement by switching to the expert. Our approach can also estimate this new metric using essentially the same tools.

## Acknowledgments and Disclosure of Funding

The authors thank Edward H. Kennedy and anonymous reviewers for their helpful comments and suggestions. AR acknowledges support from NSF grants IIS-2229881 and DMS-2310718. This work used the Extreme Science and Engineering Discovery Environment (XSEDE), which is supported by National Science Foundation grant number ACI-1548562. Specifically, it used the Bridges-2 system, which is supported by NSF award number ACI-1928147, at the Pittsburgh Supercomputing Center (PSC).

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
