# OpenReview forum: "Counterfactually Comparing Abstaining Classifiers"
_NeurIPS.cc/2023/Conference — NeurIPS 2023 poster_

### Official Review · Reviewer_9d5v · 2023-06-15

**Soundness:** 4 excellent
**Presentation:** 3 good
**Contribution:** 2 fair
**Rating:** 4
**Confidence:** 3

**Summary:**

This paper outlines a method for estimating the counterfactual performance of an abstain classifier on cases where it abstained (what would have happened if it had not abstained?). Using tools from causal inference and the potential outcomes framework in particular, they cast this is a missing data problem, and provide estimators for the relevant quantity, proving identifiability under standard conditions and showing convergence. Experimentally, they use semi-synthetic setups to show that this estimator produces good confidence intervals.

**Strengths:**

- This paper is clearly presented and technically sound
- the notion of connecting abstaining classifiers to missing data problems is interesting and I believe novel
- the demonstration of the applicability of the missing at random (conditional on X) assumption to this problem is insightful and helpful
- experiments are clear and I believe the practical effectiveness of this method

**Weaknesses:**

- Motivation: I'm not convinced that the problem this paper solves is a realistic one - the examples seem fairly contrived to me. I'm not sure why any API would be provided in the form described in 1.1 (with a free abstaining tier), and examples in the appendix all assume an  importance of the "hidden" predictions which doesn't seem super realistic. For instance, in A.3 it doesn't seem well motivated to want to evaluate a classifier's fairness based on its hidden predictions, as those do not actually result in impacts or harms on downstream populations.
- Novelty: I think that the connection of this problem to causality + missing data is novel, but I'm not sure there's much novel insight here. I do think the argument around the missing at random assumption is good (Assumption 2.1) as stated previously, but there isn't much else in this paper which is specific to the abstaining classifier problem.

Smaller notes:
line 4: "stake" -> stakes
line 50: unclear what "provably unavoidable" means
line 213: is \mu_0 defined anywhere?
line 225: the word "either" here is confusing - probably want to use "each" instead
line 324: the word "nuisance" is used many times in this section - is what it refers to defined?
line 397: might be worth defining the Gini impurity more clearly and explaining why this is a good metric


**Questions:**

- what are some other ways that missing data/causality tools can be adapted to this problem specifically? for instance, are there implications around confidence-based rejection that can be leveraged for estimation methods?
- in some cases, the underlying prediction may not even be well defined when R = 1 - for instance in cascading models where computation is only proceeded with after the rejection decision is made (if necessary). how do these models fit into the conceptualization of this paper?

---

> ### Author Rebuttal · Authors · 2023-08-10
>
> Thank you for your review. We sincerely hope that our point-by-point responses below address your concerns.
>
> ---
> > Motivation
>
> Please see our common response to all reviewers for this point.
>
> ---
> > Novelty
>
> First, highlighting and formalizing the connection between abstaining classifier evaluation and causal inference is a central contribution of our work. Given that the connection is novel, it was imperative that we first clearly establish the correspondences across the topics, such as how the MAR condition translates to having independent evaluation data, and how the positivity condition reveals a need for a policy-level approach, before jumping onto new methods.
>
> We also emphasize that, while the proof techniques are adapted from the causal inference literature, the framework and the results are new to the abstaining classifier literature, and the theorems are also not direct corollaries of any previously stated result in a different setup (e.g., for the ATE).
>
> Aside from advocating for a causal point-of-view to the abstaining classifiers literature, our work presents a shift of focus from the _learning_ perspective (how to train), which represents the view of most papers in the literature, to the _(black-box) evaluation_ perspective (how to compare), which represents many users and regulators of the classifiers. The evaluation view reveals that there are cases when we want the hidden predictions to be good nevertheless, as opposed to the hidden predictions being bad in the learning view. In App. A.2, we make this contrast explicit by comparing our score with Condessa et al. (2017)’s score, which is an evaluation metric that rewards having low accuracy on abstentions. We state there that our method can also estimate this score, as it still involves a counterfactual (of the expected score under abstentions). Note that Condessa et al.’s paper still largely takes the learning view, as their approach requires white-box access to the underlying model and assumes that 'good' abstaining classifiers would have bad performance on their abstentions.
>
> Yet another insight that we view as novel and important is in our discussion (Sec. 5 & App. D) relating the positivity condition to policy-level approaches in safety-critical contexts, in which abstaining classifiers are popularly used. The concrete suggestion we make about requiring an upper bound on the abstention rate is quite specific to abstaining classifiers.
>
> We believe that these insights add substantially to the literature on evaluating black-box abstaining classifiers, and we hope that you reconsider your assessment of the novelty of our work.
>
> ---
> > What are some other ways that missing data/causality tools can be adapted to this problem specifically?
>
> One example in our paper is to use methods that handle positivity violations, such as sample trimming. These may yield valid inference under certain deterministic abstentions, at the cost of restricting inference to a subpopulation. Other adaptable tools include diagnostics for positivity violations (Petersen et al., 2012; Lei et al., 2021) and sensitivity analysis for MAR violations (e.g., measuring how much of the evaluation data is "contaminated" by training data; see Bonvini & Kennedy, 2020). Our framework opens up these avenues for future work (noted in the revision).
>
> > for instance, are there implications around confidence-based rejection that can be leveraged for estimation methods?
>
> If a classifier makes confidence-based rejections, then the abstention mechanism $\pi$ should reflect the confidence model across inputs. This means that our estimated abstention mechanism $\hat\pi$ can reveal the confidence regions of the classifier. Of course, this is just a learned binary classifier, and it is a byproduct of our overall estimation procedure.
>
> ---
> > In some cases, the underlying prediction may not even be well defined when R = 1 - for instance in cascading models where computation is only proceeded with after the rejection decision is made (if necessary). how do these models fit into the conceptualization of this paper?
>
> The broad answer here would be that the counterfactual score can always be defined whenever a supposed prediction on the classifier’s abstentions/rejections can be meaningful to the evaluator. If there are no supposed predictions on the rejections, then the counterfactual score would not make much sense as a metric.
>
> For cascading models specifically, the counterfactual framework can assess the performance of cascading classifiers in a unique way. Imagine a two-stage cascading classifier that, in its first stage, uses a small model $f_0$ and then determines whether it is necessary to go through the expensive computation of the larger model $f_1$. Let $\pi$ be the mechanism with which the model defers its predictions. The counterfactual score of the pair $(f_0, \pi)$ can be defined straightforwardly as the expected score had only the small model been used, and it can be estimated as long as we know when the large model was invoked. For the large model, we can assess the counterfactual score of the pair $(f_1, \bar\pi)$, where $\bar\pi = 1 - \pi$, and this would correspond to the expected score had only the large model been used. These ideas can be extended to 3+ stage cascading classifiers if we are interested in estimating the score of a classifier in each stage individually.
>
> We note that the two-stage setup is relevant to the learning-to-defer setup, which Reviewer 9p41 pointed out, where the large model is an (imperfect) "expert" on the task. In such a setup, we can devise a target that combines (i) the selective score of the small model and (ii) the relative improvement by switching to the expert. As we point out in our response to R#9p41, our approach can also estimate this new target using essentially the same tools.
>
> ---
> > Smaller notes
>
> We incorporated many of these in our revised draft. We defined $\mu_0$ in line 212 and nuisance functions in line 245.

---

> > ### Comment · Reviewer_9d5v · 2023-08-11
> > **Response**
> >
> > Thanks for the rebuttal.
> >
> > - I appreciate the points in the main response re: motivation and I think that the more fully fleshed out argument is more compelling than what's in the paper currently. I think it's probably important to include a more in-depth motivation section in a paper like this.
> > - I agree that there is novelty in the abstaining classifier-missing data connection and it's possible I underrated it's value in my original review.
> > - I agree that the counterfactual score is defined in the cascading case - my fault for stating it poorly in the original review.
> >
> > In general, on re-reading my review and scanning the paper again, I'm inclined to re-consider my score, and will do so as I discuss with the other reviewers.

---

> > > ### Author Response · Authors · 2023-08-11
> > > **Author Response**
> > >
> > > Thank you for your encouraging response. We will further flesh out the motivation section of the paper by incorporating the additional points from our rebuttal.
> > >
> > > We also appreciate you for bringing up the cascading model example, and we will incorporate our response in the paper.
> > >
> > > We hope that these responses help you reconsider your assessment of our work.

---

### Official Review · Reviewer_9p41 · 2023-07-05

**Soundness:** 3 good
**Presentation:** 3 good
**Contribution:** 2 fair
**Rating:** 6
**Confidence:** 3

**Summary:**

This paper proposes to compare black-box abstaining classifiers (neither the base classifier nor the abstention mechanism is known to the evaluator) by the counterfactual score, that is the expected score of the abstaining classifier had it not been given the option to abstain. They prove that this quantity is identifiable under two standard conditions, that are missing at random and positivity, and then develop nonparametric and doubly robust methods to estimate it. Experiments on simulated data and CIFAR-100 are conducted.

**Strengths:**

This paper is well-organized and clearly written. Comparing abstaining classifiers is an important problem in machine learning and the counterfactual score proposed in this paper is a new evaluation metric that is relevant in practice.

**Weaknesses:**

1. As stated in the paper, the counterfactual score is unidentifiable if abstentions are deterministic which includes a fair amount of abstention methods in the literature.
2. It would be better to briefly discuss some related works on abstention instead of just referring to surveys.

**Questions:**

1. In the learning-to-defer setting, where instead of rejecting an example, the example can be deferred to an expert and let that expert make the prediction, does the counterfactual score still make sense?

minor:
1. There is a redundant 'is' at the end of line 420.

**Limitations:**

Yes.

---

> ### Author Rebuttal · Authors · 2023-08-10
>
> Thank you for your feedback on our work and for acknowledging the practical importance of our work and the problem. See below for our responses to each of your concerns and questions.
>
> ----
>
> > As stated in the paper, the counterfactual score is unidentifiable if abstentions are deterministic which includes a fair amount of abstention methods in the literature.
>
> While we acknowledge the concern from an applicability standpoint, we emphasize that this should not be considered a weakness from a methodology standpoint, given that the score is never identifiable under deterministic abstentions (lines 194–196). As we state in lines 424–428, if we are interested in the counterfactual score of an abstaining classifier, then no other method can estimate it under deterministic abstentions without resorting to restrictive modeling assumptions. This is why we first mention a policy-level approach that may require vendors to meet the positivity requirement in order to enable buyers to compare the products before choosing one (lines 419–423 and Appendix D).
>
> In our opinion, the fact that our paper does not apply to all abstaining classifiers is perhaps ok, as long as we provide a solution for stochastically abstaining ones and we are upfront about this restriction. We’d also like to reiterate that there are recent papers that show the superiority of stochastically abstaining classifiers (lines 201–210), similar to the effectiveness of randomized classifiers in the fairness literature.  We believe that the counterfactual framework can be more relevant in the future when more abstaining classifiers adopt stochastic mechanisms.
>
> ----
>
> > It would be better to briefly discuss some related works on abstention instead of just referring to surveys.
>
> We have now added a few more references and details, but given that our approach is largely agnostic to the specific learning algorithms for abstaining classifiers, we put our emphasis on the evaluation side. We do explicitly mention directly relevant papers on abstention throughout the paper, including Chow (1970) and El-Yaniv and Wiener (2010) for their discussion of the selective score & coverage formulations, as well as the recent works that develop methods using stochastic abstentions (Kalai and Kanade, 2021; Schreuder and Chzhen, 2021).
>
> ----
>
> > In the learning-to-defer setting, where instead of rejecting an example, the example can be deferred to an expert and let that expert make the prediction, does the counterfactual score still make sense?
>
> Yes, the counterfactual score makes sense. In the learning-to-defer setting involving an expert, the counterfactual score would refer to the expected score of the overall system had the classifier not deferred at all. The counterfactual score is thus an evaluation metric primarily for the classifier, and it is independent of the expert’s predictions, even when the classifier is adaptive to the expert’s tendencies.
>
> In the case where the goal is to assess the _joint_ performance of the algorithm and the expert, then it may be useful to estimate a variation of Condessa et al. (2017)’s score, which we summarize in both Section 2 and Appendix A.2. If we denote the expert’s score as $E$, then equation (1) in the Appendix can further be generalized to
> $$
> \theta^E := \mathbb{E}[S \mid R=0] \mathbb{P}(R=0) + \mathbb{E}[E - S \mid R=1] \mathbb{P}(R=1).
> $$
>
> For each rejection ($R=1$), we assess the system by the difference in the quality of expert prediction and the model prediction ($E-S$). If the expert is an oracle ($E=1$), then this recovers Condessa et al.’s score. Note that Condessa et al. primarily focused on defining and justifying $\theta$ in the “white-box” setting, where the score on rejections is known, and it does not discuss the counterfactual estimation problem that arises in the black-box setting.
>
> When it comes to _estimation_, as we state in Appendix A.2, our method can analogously estimate Condessa et al.’s score $\theta$. Thus, the approach can further estimate $\theta^E$, whenever we can estimate $\mathbb{E}[E \mid R=1]$. If we are estimating the performance of a joint system, possibly learned using a learning-to-defer method, then we can estimate $\theta^E$, as long as either the expert’s performance is known (e.g., $E=1$ for an oracle) or the deferral decision $R$ is recorded.

---

> > ### Comment · Reviewer_9p41 · 2023-08-17
> >
> > Thank you for your response. It addressed some of my concerns. I have also read the other reviews and rebuttals and would like to keep my initial rating.

---

### Official Review · Reviewer_fENC · 2023-07-05

**Soundness:** 3 good
**Presentation:** 4 excellent
**Contribution:** 3 good
**Rating:** 7
**Confidence:** 3

**Summary:**

The authors focus on the problem of evaluating abstaining classifiers while also taking account their counterfactual predictions had they not abstained on certain data points. As an evaluation metric they consider the expected accuracy of the classifiers over both their actual and counterfactual predictions, which they define as the counterfactual score. They provide the conditions under which the counterfactual score is identifiable and also propose a doubly robust estimator to compute the counterfactual score of an abstaining classifier. Finally, they evaluate their estimator with simulations and experiments on  real data.

**Strengths:**

The idea of considering the counterfactual predictions of the classifier had it not abstained for evaluation is conceptually quite interesting and also could have a significant impact in scenarios when the expert, for example,  who predicts when the classifier abstains is also uncertain and would still choose the classifier’s prediction.  The paper is very nicely written, well structured and organised, well motivated and convincing. The experimental evaluation includes both simulations and real data experiments, the setup and the results are clearly explained and also the code is provided for reproducibility.

**Weaknesses:**

There see to be in general no major weaknesses. It is a bit confusing the pointer to table 1 in line 86, where the notation has not yet been introduced. It would have been nicer perhaps not to include notation at that point or explain the notation in the caption. Moreover, the captions of the tables should have been above the tables according to the author instructions.
Typos:

-Line  87: ‘of significant interest’

-Line 367: there seem to be a missing verb

-Line 420: the sentence does not make sense, it seems that some part is missing.

**Questions:**

N/A.

**Limitations:**

The authors adequately address the limitations of their work.

---

> ### Author Rebuttal · Authors · 2023-08-10
>
> We sincerely appreciate your positive and comprehensive feedback on our work.
>
> In our revision, we incorporated all of your editorial notes, including moving Table 1 to the beginning of Section 2 and having all table captions appear before the tables themselves.

---

### Official Review · Reviewer_paQd · 2023-07-06

**Soundness:** 3 good
**Presentation:** 2 fair
**Contribution:** 2 fair
**Rating:** 5
**Confidence:** 4

**Summary:**

The paper proposes a new way to evaluate counterfactually, abstain classifiers. By considering the task of selective classification, the authors reformulate the problem into a Missing data problem which can thus be seen as a causal inference (Counterfactual) problem.
The authors also state the corresponding assumptions needed for the above to be identifiable and propose a doubly robust estimator for their proposed metric.

**Strengths:**

- The paper proposes a new view on evaluation of abstain classifiers through the lens of Rubins Counterfactuals
- The paper clearly states the assumptions for the problem to be identifiable
- The paper clearly formulates the approach of doubly robust estimations and how it can be used in their setting for abstain classifier comparisons.


**Weaknesses:**

- My biggest concern of the paper is the motivation. As stated in their example 1.1 the example they have given is quite contrived and not really applicable in most cases. I believe much stronger motivating examples would be useful. If the authors would clearly give me better examples (in addition to the appendix ones) that would greatly help me position this paper.
- Secondly, I believe the experiments are not clearly written out to me. Could the authors tell me how exactly this metric is preferred over simply looking at the accuracy of the selected labels and the inaccuracy of the non-selected labels? I might have misunderstood this part, so if the authors could help me clarify this part it would be very helpful. The authors mention that this is an "inverse" problem, but not sure if I understood this part.
- Lastly, I wonder if the authors have thought of implementing the second point mentioned in this review. I am confused about why there are no other baselines. I would have thought that the second point would constitute a simple baseline.

**Questions:**

see in weakness
I am more than happy to raise my score if the above are clarified.

**Limitations:**

Yes the authors have addressed limitations of their work

---

> ### Author Rebuttal · Authors · 2023-08-10
>
> Thank you for your review of our work and for acknowledging the novelty and clarity of our exposition. See below for our responses to each of your concerns.
>
> ----
>
> > My biggest concern of the paper is the motivation. As stated in their example 1.1 the example they have given is quite contrived and not really applicable in most cases. I believe much stronger motivating examples would be useful. If the authors would clearly give me better examples (in addition to the appendix ones) that would greatly help me position this paper.
>
> Please see our common response to all reviewers which includes our response to this point.
>
> ----
>
> > Secondly, I believe the experiments are not clearly written out to me. Could the authors tell me how exactly this metric is preferred over simply looking at the accuracy of the selected labels and the inaccuracy of the non-selected labels? I might have misunderstood this part, so if the authors could help me clarify this part it would be very helpful. The authors mention that this is an "inverse" problem, but not sure if I understood this part.
>
> In brief, if two classifiers make predictions on different subsets of the data, have different frequencies and patterns of abstaining, and have different accuracies on the ones they do predict, it is actually not so obvious how to combine that information to compare them in a sensible and fair manner.
>
> Our examples (1.1 and A.1-A.3) motivate cases in which it _hurts_ to be inaccurate on abstentions (non-selected labels):
> * If a free-trial classifier from Example 1.1 is highly inaccurate in its abstentions, and we (the evaluator) are impressed by its performance on non-abstentions and purchase the full non-abstaining service, then we will suffer a decrease in the total accuracy from the free trial phase to the paid use phase.
> * If the self-driving car from Ex. A.1 or the hospital from Ex. A.2 attempts to use the abstaining classifiers’ hidden predictions in a failure mode, then it will suffer more when the classifier is less accurate on its abstentions.
> This is the “inverse” sense from a typical _learning_ scenario for abstaining classifiers: whereas learning objectives for abstaining classifiers often reward abstaining on inaccurate predictions, the counterfactual score is adequate for evaluation setups where such abstentions can be hurtful.
>
> The metric that you discuss here roughly corresponds to Condessa et al. (2017)’s score, which we formally defined in Appendix A.2. We state there that:
> * The two metrics assess abstaining classifiers from different viewpoints, depending on whether it is good or bad for a classifier to perform poorly on its abstentions.
> * Our estimation approach can be applied to estimate Condessa et al.’s score in the black-box setting, in which their score is also a counterfactual (we do not know how well or poorly the classifier performs on its abstentions). Note that Condessa et al. primarily focused on defining their score in the “white-box” setting, in which the classifier’s accuracy on its abstentions is known, but we do not assume this.
>
> Finally, as for the experiments, given that our primary method is a statistical inference approach (a confidence interval), we focus on empirically examining its validity/coverage, that is, whether a 95% CI correctly covers the true parameter (fully known in simulated experiments) approximately 95% of the time across repeated simulations. The complementary metric to coverage is the power, as measured by how tight the CI is (a tighter CI would give more certainty to the user). Our real data experiment validates how the CI estimates a number that we expect to see in each scenario (zero if the base classifiers were the same; nonzero otherwise). These experiments are NOT meant to provide further justifications on why the counterfactual score is a viable alternative _as a metric_ to existing ones; they are simply meant to show that our method for estimating the counterfactual score is sensible and works well in practice.
>
> In our revision, we clarified the main goals of our experiments up front (at the beginning of the section).
>
> ----
>
> > Lastly, I wonder if the authors have thought of implementing the second point mentioned in this review. I am confused about why there are no other baselines. I would have thought that the second point would constitute a simple baseline.
>
> As per our previous response, our primary baselines are other _estimation_ approaches, namely the plug-in and IPW estimators, and not other _metrics_. The metrics can be computed straightforwardly, but they do not reveal insights about whether our methods work with simulated and real data, and it is hard to compare different methods that are meant for different metrics.
>
> Having said that, we already include the most commonly used metric for evaluating abstaining classifiers: a combination of the selective score and coverage. In our simulated experiments, these numbers are shown in the first paragraph of Sec. 4.1. Note that both of these quantities are just sample averages, and a standard CI can be computed for estimation. Another existing metric would be Condessa et al.’s score, which we recap in App. A.2 and in the above. For our simulated setup, this can also be computed straightforwardly: classifier A obtains 0.4145, while classifier B obtains 0.4345 (they both abstain quite a bit, and A is heavily penalized for abstaining on correct predictions).
>
> ----
>
> We sincerely hope that our responses address your concerns.

---

> > ### Comment · Reviewer_paQd · 2023-08-18
> > **Thanks for the reply**
> >
> > First of all i would like to thank the authors for the reply.
> >
> > The motivation part has been partially justified for me, even though i remain skeptical, I believe there is a chance this might become useful in the future.
> >
> > In terms of the metric, I do not understand exactly why the metric by Condessa et al is not applicable in "evaluating" abstain classifier? To me it seems like the most natural and basic way to evaluate a selective classifier. Could the authors please let me know what the proposed score does, that current methods cant exactly. I would like to hear an exact and precise example please.
> >
> > Also I dont understand this sentence. So what is the point then? I am really confused now, could you please clarify this part. "These experiments are NOT meant to provide further justifications on why the counterfactual score is a viable alternative as a metric to existing ones; they are simply meant to show that our method for estimating the counterfactual score is sensible and works well in practice".
> >
> > As it stands, i cannot raise my score and hope that the authors can convince me otherwise.

---

> > > ### Author Response · Authors · 2023-08-19
> > > **Further clarification**
> > >
> > > > In terms of the metric, I do not understand exactly why the metric by Condessa et al is not applicable in "evaluating" abstain classifier? To me it seems like the most natural and basic way to evaluate a selective classifier. Could the authors please let me know what the proposed score does, that current methods cant exactly.
> > >
> > > We did not say anywhere that Condessa et al.’s metric is not applicable in evaluating an abstaining classifier. We elaborate on when the counterfactual score is preferred over Condessa et al.’s score in lines 89–103 in our paper, including the following line:
> > > _While [Condessa et al.’s] view is relevant to the training of abstention rules, it is at odds with black-box settings where the underlying predictions may still be executed even when the method abstains, motivating the counterfactual score._
> > >
> > > The questions of (1) whether a score is natural and (2) what “current methods can’t do” are separate.
> > >
> > > 1. Our whole intro is devoted to justifying that the counterfactual score, as we defined it in our paper, can be useful (rather than Condessa et al.’s). Both metrics can be viable metrics for evaluating abstaining classifiers, but they simply evaluate the classifiers in a different manner. The two scores can be useful in different scenarios, and we focused on scenarios in which our proposed metric can be useful.
> > > 2. Condessa et al. (or any other work, to our knowledge) do not discuss the black-box setting that requires a _statistical estimation_ approach, as they already assume full knowledge of whether the classifier would have classified each input correctly or not. But this is not known in any of the black-box settings we describe. Both Condessa et al.’s score and the counterfactual score require how accurate the classifier _would have been_ on its abstentions, and we do not know this in the black-box scenario like Example 1.1 (we have no clue about whether an API would get a 100% or 0% accuracy on inputs that it chose to abstain from making predictions). This unknown quantity is really the “counterfactual,” and thus we devote Sections 3 and 4 of our paper to discuss how we can _estimate_ the counterfactual score in the black-box setting. No other "current method" can estimate the counterfactual score (because it was never defined formally), and we further mention in Appendix A.3 that our method can further estimate Condessa et al.’s score in the black-box setting.
> > >
> > > > I would like to hear an exact and precise example please.
> > >
> > > To give a simplified example, suppose that we compare two abstaining classifiers on 100 data points, whose inputs are sampled uniformly on $[-1, 1] \times [-1, 1]$. Suppose that classifier A achieves a 1.0 accuracy on the left half of the input space ($x_1 < 0$) but a 0.8 accuracy on the right half ($x_1 \geq 0$). It abstains at an 80% rate on the right half, while it does not abstain at all on the left half. On the left half, the classifier makes 50/50 correct predictions on the left half; on the right half, it makes 8/10 correct predictions and 40 abstentions (for which it would have been correct 80% of the time).
> > >
> > > Recall the definitions from Appendix A.3. The counterfactual score of classifier A is 58/60 * 0.6 + 32/40 * 0.4 = 0.9, whereas Condessa et al.’s score for this classifier is 58/60 * 0.6 + (1 - 32/40) * 0.4 = 0.66. Note that the classifier’s selective score is 58/60=0.97 while its coverage is 60/100=0.6.
> > >
> > > Next, suppose that classifier B is the same as classifier A, except that it achieves a 0.6 accuracy on the right half. Then, classifier B’s counterfactual score would be 0.8, lower than classifier A, whereas Condessa et al.’s score for B would be 0.82.
> > >
> > > If the evaluator needs to access the classifier’s hidden predictions on the right half, they would prefer classifier A, which has a higher accuracy had no abstentions were made. In Example 1.1, the right half would correspond to the inputs that the free-trial API chose to abstain on.
> > >
> > > > Also I dont understand this sentence. So what is the point then? I am really confused now, could you please clarify this part. "These experiments are NOT meant to provide further justifications on why the counterfactual score is a viable alternative as a metric to existing ones; they are simply meant to show that our method for estimating the counterfactual score is sensible and works well in practice".
> > >
> > > Because we developed an estimation approach, we need experiments to validate whether the estimator actually estimates the target quantity well on simulated (and real) data. We note that it is _not_ trivial whether this score can be estimated well in practice, given that it is an unknown counterfactual quantity. The real data experiments illustrate different comparison scenarios and what the estimated counterfactual score would be.
> > >
> > > We hope that these responses clarify your concerns.

---

### Author Rebuttal · Authors · 2023-08-10

We appreciate all of our reviewers for their feedback on our work. We are particularly thankful to them for acknowledging the problem’s significance as well as the novelty of the connections we elucidate between abstaining classifiers, black-box evaluation, and causal inference.

A main concern shared by Reviewers paQd and 9d5v is about the practicality of our motivating examples, so we give a common response here. Given the fairly recent advances in learning algorithms for abstaining classifiers, we find it imperative to lay out the tools with which we can evaluate and compare the resulting models in imaginable settings. We argue in the paper that existing metrics, such as Chow’s score and other combinations of selective score and coverage, are not adequate in many such scenarios. Thus, we take the first step in a different type of formal evaluation of black-box abstaining classifiers, by taking a counterfactual point of view.

When it comes to practicality, we believe that not all works need to be directly motivated by an existing practical problem. One can foreshadow conceptual problems and work to address them in advance of them showing up in practice. As abstaining classifiers become more common, so will the desire to compare them, and especially ask counterfactual questions about their performance had they not abstained.

We specifically elaborate on two of our examples, in response to Reviewer 9d5v.
* Ex. 1.1: Closed-source ML APIs, like those offered by Azure, AWS, Google, and many startups, are becoming ever more popular, and the predictions they provide are clearly important assets of theirs. It is not difficult to imagine that these services will deem certain predictions as more valuable than others (think of classifying CT scans). This can incentivize them to allow users to try a free version of the software that abstains, such that potential buyers can get some sense of the quality of predictions but yet avoid giving away all valuable predictions.
* Ex. A.3: This is one example in which the hidden predictions are not directly used, but it rather illustrates a different use case in which the hidden predictions may explain the inner workings of an otherwise black-box prediction model. For example, the auditor may notice that certain demographic groups receive more abstentions by the model than others. In that case, the counterfactual score may reveal the extent to which the predictions on those groups are bad. If they are actually much worse, then better training data or fairer classifiers may be needed; if not, then only a fairer abstaining mechanism may be needed.

In response to Reviewer paQd: we think that the clarifications above may help address the motivation aspect more clearly than coming up with additional examples. Nevertheless, we are happy to discuss further during the discussion period about these and other examples.

Aside from the motivation, the reviewers also raised interesting questions about our paper, including connections to the learning-to-defer settings and cascading models. Please see our point-by-point responses to each reviewer’s comments below.

---

### Decision · Program_Chairs · 2023-09-21

**Decision:**

Accept (poster)

**Comment:**

The authors introduce a method based on counterfactual inference to estimate the (counterfactual) performance of "abstaining" classifiers on sample where it has abstained. A majority of the reviewers are positive about the paper and think it should be accepted.